# Functional tissue units in the Human Reference Atlas

Supriya Bidanta[1,5], Katy Börner [1,5] ✉, Bruce W. Herr II [1], Ellen M. Quardokus[1], Marcell Nagy [2,3], Katherine S. Gustilo[1], Rachel Bajema[1], Elizabeth Maier[1], Roland Molontay [2,3] & Griffin M. Weber[4] ✉

Functional tissue units form the basic building blocks of organs and are important for understanding and modeling the healthy physiological function of the organ and changes that occur during disease states. In this comprehensive catalog of 22 anatomically based, nested functional tissue units from 10 healthy human organs, we document the definition, physical dimensions, blood vasculature connections, and cellular composition. All anatomy terms are mapped to the multi-species Uber-anatomy Ontology (Uberon) and cells are mapped to Cell Ontology to support computational access via standardized metadata. The catalog includes datasets, illustrations, and a large printable poster illustrating how the blood vasculature connects the 22 functional tissue units in 10 organs. All data and code are freely available. The work is part of an ongoing international effort to construct a Human Reference Atlas of the 37 trillion cells that make up the healthy human body.

The Human Reference Atlas (HRA) is an international effort of 17 consortia to build a freely available map of the healthy (i.e., non-diseased) adult human body down to the single-cell level. At the highest level are organ systems, whose individual organs contain various tissues, also called anatomical structures. Tissues, in turn, consist of repeating structures known as functional tissue units (FTUs).

Bernard de Bono et al. define FTUs as "a three-dimensional block of cells centered around a capillary, such that each cell in this block is within diffusion distance from any other cell in the same block"[1]. The description of the cells within an FTU and the anatomical location of the FTU form a so-called primary tissue motif (PTM). An FTU must support both metabolism and communication between cells. The maximum diffusion distance of oxygen and other molecules constrains the size of an FTU. Therefore, tissues and organs in the body consist of many repeating FTUs to perform the overall function of the organ.

We previously revised this original definition to meet the needs of the HRA effort by describing an FTU as "the smallest tissue organization that performs a unique physiologic function and is replicated multiple times in a whole organ"[2]. Here, we extend this definition of FTUs by allowing them to form a nested hierarchy. This is needed in cases where multiple FTUs work together to accomplish complex functions that the individual FTUs cannot perform on their own. Just like the individual FTUs, these higher-level groups of FTUs are replicated many times.

This research article builds on and extends the Human Reference Atlas (HRA) effort[3]. Specifically, we present a catalog of the first 22 FTUs for the HRA, including the kidney nephron as an example of a nested FTU made of smaller FTU structures. The current version of this catalog contains (1) the length and diameter of each FTU; (2) the blood vasculature pathways that connect FTUs to the heart and between each other; (3) the cell types within each FTU; and (4) 2D illustrations of each FTU showing their prototypical cell types, cell shapes, sizes, and spatial arrangement. We map all anatomical structures to their corresponding Uber-anatomy Ontology (Uberon) ID and all cell types to their Cell Ontology (CL) ID[4] in a crosswalk table to enable linkage to other datasets. We reference supporting scholarly paper evidence where applicable. The complete catalog of FTUs is available as machine-

[1]Department of Intelligent Systems Engineering, Luddy School of Informatics, Computing, and Engineering, Indiana University, Bloomington, IN 47408, USA. [2]Department of Stochastics, Institute of Mathematics, Budapest University of Technology and Economics, H-1111 Budapest, Hungary. [3]Institute of Biostatistics and Network Science, Semmelweis University, H-1085 Budapest, Hungary. [4]Department of Biomedical Informatics, Harvard Medical School, Boston, MA 02115, USA. [5]These authors contributed equally: Supriya Bidanta, Katy Börner. ✉e-mail: katy@iu.edu; griffin_weber@hms.harvard.edu

readable downloadable files via the HRA Portal (https://humanatlas.io/2d-ftu-illustrations) and as a large printable poster.

Much of the information we present here about FTUs exists in the literature but is siloed by organ or tissue type, and no unifying conceptual or data schema framework exists. The creation of a catalog of FTUs as an integral part of the HRA, along with standardized descriptions linked to ontologies, publications, and experimental data, makes it possible to compare and contrast these units throughout the human body. This way, we can gain a better comprehension of how FTUs function together and how they can be integrated into models and simulations.

## Results

### Nested organization of FTUs
The 6th release of the HRA v2.0 covers 22 FTUs distributed across 10 different organs, see Table 1. Top-level FTUs are composed of smaller FTUs (e.g. the kidney nephron is the top-level FTU composed of seven nested FTUs). The 15 top-level FTUs by organ are (1) kidney: nephron; (2) large intestine: crypt of Lieberkuhn of colon; (3) liver: liver lobule; (4) lung: alveolus and bronchus submucosal gland; (5) pancreas: islets of Langerhans, pancreas acini, and intercalated duct; (6) prostate gland: prostate glandular acinus; (7) skin: dermal papilla and epidermal ridge of digit; (8) small intestine: intestinal villus; (9) spleen: red pulp and white pulp; and (10) thymus: thymus lobule. The kidney nephron contains seven smaller FTUs: renal corpuscle, inner medullary collecting duct, descending limb of loop of Henle, loop of Henle ascending limb thin segment, thick ascending limb of loop of Henle, outer medullary collecting duct, and cortical collecting duct. Each FTU component of the larger nephron FTU performs a specific independent filtering step, but all are needed to perform the overall function of the nephron. A 2D-FTU crosswalk table links all anatomical structures and cell types in the 22 FTUs to the Uberon and Cell Ontology, respectively.

### Geometric properties of FTUs
Table 1 lists the 10 organs and their 22 FTUs together with the Uberon ID, spatial dimensions, and reference paper(s) in which the dimensions were published. For spherically-shaped FTUs, such as the *alveolus of lung*, a single dimension value (or range) is listed representing the typical diameter of the FTU. For cylindrical FTUs, such as the *inner medullary collecting duct* in the kidney, length and diameter are provided. The diameter of the *intercalated duct* is the largest in the head of the pancreas and smallest in the tail.

### Blood vasculature connections to FTUs
As part of the HRA initiative, we previously published a comprehensive database of nearly one thousand blood vessels throughout the human body, which includes information about their branching structure, cell types, biomarkers, and other properties[5,6]. We call this the HRA Vasculature Common Coordinate Framework (HRA-VCCF) database. Because blood vessels extend to all parts of the body, the location of cells can be described relative to nearby vessels. This creates a kind of intrinsic coordinate system within the body. Table 2 lists the vessels in the HRA-VCCF that directly supply or drain each FTU. Note that the epidermal ridge of digit does not contain blood vessels but rather obtains oxygen via diffusion from the underlying dermal papilla. A list of the full vasculature pathways from the heart to each FTU and back to the heart can be found in Supplementary Data 1.

Where possible, vessels are mapped to their corresponding Uberon ID or Foundational Model of Anatomy (FMA) ontology IDs[7]. About 83% of the vessels exist in one or both of these ontologies when considering Uberon and/or FMA[5].

### Cell Types
A list with the different types of cells in each of the 22 FTUs is available at the HRA Portal (https://humanatlas.io/2d-ftu-illustrations). Table 3 shows a subset of this list for FTUs within the kidney nephron. Each cell type is associated with its corresponding Cell Ontology (CL) ID.

**Table 1 | Functional tissue units by organ, with Uberon ID, and dimensions (length where applicable and diameter in millimeters) as listed in the provided references**

| Organ | FTU | Uberon ID | Dimensions (mm) | References |
|---|---|---|---|---|
| Kidney | Nephron | UBERON:0001285 | 30–50 | 13 |
| Kidney | –Renal corpuscle | UBERON:0001229 | 0.15–0.25 | 14 |
| Kidney | –Inner medullary collecting duct | UBERON:0004205 | 12 × 0.05 | 15 |
| Kidney | –Descending limb of loop of Henle | UBERON:0001289 | 3.2 × 0.026 | 15 |
| Kidney | –Loop of Henle ascending limb thin segment | UBERON:0004193 | 5 × 0.026 | 15 |
| Kidney | –Thick ascending limb of loop of Henle | UBERON:0001291 | 5 × 0.026 | 15 |
| Kidney | –Outer medullary collecting duct | UBERON:0004204 | 5 × 0.040 | 15 |
| Kidney | –Cortical collecting duct | UBERON:0004203 | 20–22 × 0.02–0.05 | 15,16 |
| Large intestine | Crypt of Lieberkuhn | UBERON:0001984 | 0.05 × 0.7 | 17 |
| Liver | Liver lobule | UBERON:0004647 | 1–2.5 | 18,19 |
| Lung | Alveolus of lung | UBERON:0002299 | 0.184–0.2 | 20 |
| Lung | Bronchus submucosal gland | UBERON:8410043 | 1–2 | 21 |
| Pancreas | Islets of Langerhans | UBERON:0000006 | 0.1 | 22 |
| Pancreas | Pancreas acinus | UBERON:0001263 | 0.010–0.024 | 23 |
| Pancreas | Intercalated duct | UBERON:0014726 | 1.5–3.5 | 24 |
| Prostate gland | Prostate glandular acinus | UBERON:0004179 | 0.5 | 25 |
| Skin | Dermal papilla | UBERON:0000412 | 0.1–1.5 | 26 |
| Skin | Epidermal ridge of digit | UBERON:0013487 | 0.25–0.93 | 26–29 |
| Small intestine | Intestinal villus | UBERON:0001213 | 0.5–1.0 | 30,31 |
| Spleen | Red pulp of spleen | UBERON:0001250 | 0.02–0.04 | 32 |
| Spleen | White pulp of spleen | UBERON:0001959 | 0.5–1 | 32 |
| Thymus | Thymus lobule | UBERON:0002125 | 0.5–2 | 33–35 |

**Table 2 | Blood vessels that directly supply or drain each FTU**

| Organ | FTU | Vessels |
|---|---|---|
| Kidney | Nephron | Ascending vasa recta of kidney; descending vasa recta of kidney; glomerular capillary; peritubular capillary; renal afferent arteriole; renal efferent arteriole; vasa recta of kidney |
| Kidney | –Cortical collecting duct | Peritubular capillary |
| Kidney | –Descending limb of loop of Henle | Ascending vasa recta of kidney |
| Kidney | –Inner medullary collecting duct | Ascending vasa recta of kidney; descending vasa recta of kidney |
| Kidney | –Loop of Henle ascending limb thin segment | Vasa recta of kidney |
| Kidney | –Outer medullary collecting duct | Ascending vasa recta of kidney; descending vasa recta of kidney |
| Kidney | –Renal corpuscle | Glomerular capillary; renal afferent arteriole; renal efferent arteriole |
| Kidney | –Thick ascending limb of loop of Henle | Peritubular capillary |
| Large intestine | Crypt of Lieberkuhn | Branch of mucous plexus of colon |
| Liver | Liver lobule | Central vein of liver; hepatic arteriole; hepatic portal venule; hepatic sinusoid |
| Lung | Alveolus of lung | Alveolar capillary |
| Lung | Bronchus submucosal gland | Bronchial capillary |
| Pancreas | Intercalated duct | Periductal capillary of intercalated duct of pancreas |
| Pancreas | Islet of Langerhans | Pancreatic islet capillary |
| Pancreas | Pancreatic acinus | Intralobular arteriole of pancreas; lobular capillary of pancreas |
| Prostate gland | Prostate glandular acinus | Prostatic capillary |
| Skin | Dermal papilla | Branch of subpapillary plexus |
| Skin | Epidermal ridge of digit | Indirectly via branch of subpapillary plexus in the dermal papilla |
| Small intestine | Intestinal villus | Branch of mucous plexus of small intestine |
| Spleen | Red pulp of spleen | Post-sheath open capillary of spleen; red pulp venule of spleen; splenic venous sinusoid |
| spleen | White pulp of spleen | Branch of superficial white pulp capillary of spleen; secondary follicle arteriole of spleen; spleen central arteriole |
| Thymus | Thymus lobule | Thymic capillary |

**Table 3 | Cell types with CL IDs in kidney nephron FTUs**

| FTU | Cell Type | CL ID |
|---|---|---|
| Renal corpuscle | Parietal epithelial cell | CL:1000452 |
| Renal corpuscle | Podocyte | CL:0000653 |
| Renal corpuscle | Glomerular capillary endothelial cell | CL:1001005 |
| Renal corpuscle | Glomerular mesangial cell | CL:1000742 |
| Renal corpuscle | Epithelial cell of proximal tubule | CL:0002306 |
| Renal corpuscle | Macula densa epithelial cell | CL:1000850 |
| Renal corpuscle | Kidney afferent arteriole endothelial cell | CL:1001096 |
| Renal corpuscle | Kidney efferent arteriole endothelial cell | CL:1001099 |
| Cortical collecting duct | Kidney cortex collecting duct principal cell | CL:1000714 |
| Cortical collecting duct | Kidney connecting tubule alpha-intercalated cell | CL:4030020 |
| Cortical collecting duct | Kidney connecting tubule beta-intercalated cell | CL:4030021 |
| Cortical collecting duct | Peritubular capillary endothelial cell | CL:1001033 |
| Inner medullary collecting duct | Kidney inner medulla collecting duct principal cell | CL:1000718 |
| Inner medullary collecting duct | Peritubular capillary endothelial cell | CL:1001033 |
| Outer medullary collecting duct | Kidney outer medulla collecting duct intercalated cell | CL:1000717 |
| Outer medullary collecting duct | Peritubular capillary endothelial cell | CL:1001033 |
| Outer medullary collecting duct | Kidney outer medulla collecting duct principal cell | CL:1000716 |
| Thick ascending limb of loop of Henle | Kidney loop of Henle thick ascending limb epithelial cell | CL:1001106 |
| Thick ascending limb of loop of Henle | Peritubular capillary endothelial cell | CL:1001033 |
| Descending limb of loop of Henle | Kidney loop of Henle thin descending limb epithelial cell | CL:1001111 |
| Descending limb of loop of Henle | Vasa recta descending limb | CL:1001285 |
| Loop of Henle ascending limb thin segment | Kidney loop of Henle thin ascending limb epithelial cell | CL:1001107 |
| Loop of Henle ascending limb thin segment | Vasa recta ascending limb | CL:1001131 |

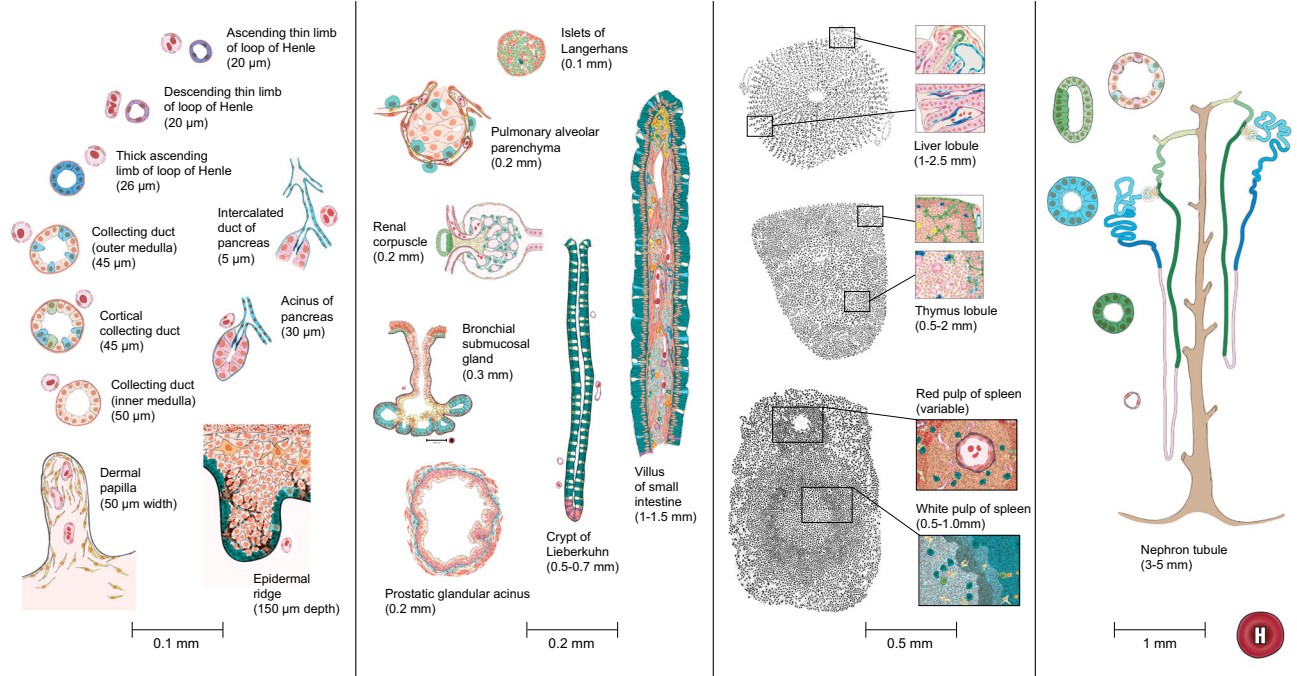

**Fig. 1 | 2D illustrations of all 22 FTUs with labels and corresponding scale bars.** The leftmost panel depicts 10 FTUs with scale bars within the 0.1 mm range, including various sections of the nephron and skin structures. The middle-left panel shows 7 FTUs of scale 0.2 mm, such as pulmonary alveoli, renal glomeruli, and pancreatic structures. The middle-right panel provides detailed views of FTUs at a 0.5 mm scale, including the liver lobule and thymus lobule. The rightmost panel shows an illustration of nephron tubules at a 1 mm scale. Scale bars, ranging from 0.1 mm to 1 mm, offer a reference for the relative sizes of each FTU.

## Spatial arrangement of cells

The spatial arrangement of cells and their types in each FTU is recorded in 2D illustrations, see https://humanatlas.io/2d-ftu-illustrations. The illustrations are saved in SVG, PNG, and AI format. Then, the SVG files are converted to JSON files with selectable structures linked to the metadata. Finally, a crosswalk table is compiled that associates 2D anatomical structures and cell types in the FTUs with their proper terms and IDs in the HRA. Illustrations of all 22 FTUs are shown in Fig. 1 at four levels of magnification, see scale bars.

## Poster-sized visualization of the FTU catalog

To place the FTUs within the larger context of the HRA, we created a visualization of all 1607 anatomical structures and 1943 cell types with their blood vasculature connections from the 5th release of the HRA. Note that there are 54 nodes for the 22 FTUs. This is because some FTUs overlap multiple anatomical structures and therefore appear in more than one location of the partonomy. The visualization is composed of two radial tree graphs: The first graph contains the nested "partonomy" of the anatomical structures and cell types in the HRA. The human body serves as the root node in the center, the largest anatomical structures (organs) are placed further out, and smaller substructures and tissue types branch outwards from the organs' leaf nodes denoting cell types. The second graph contains all the blood vessels in the HRA, with the chambers of the heart in the center, and increasing smaller vessels more distal to the heart, again branching outwards from the center. Nodes in the two radial tree graphs meet at points where the HRA indicates a vessel supplies or drains the corresponding anatomical structure. Nodes that represent FTUs are highlighted in green. The left side shows the nested partonomy and blood vessels in females, and the right side shows them for males with a "butterfly-like" appearance that invites closer examination, discussion, and self-portrait photos, see Fig. 2. The visualization can be downloaded as an Adobe Illustrator file format (AI) or as a ready-to-print PDF file format for a 6-foot (26" or 183 cm) diameter poster from https://

github.com/cns-iu/hra-ftu-vccf-supporting-information. We plan to update this poster with future releases of the HRA to reflect the ongoing development of the partonomy and vasculature graphs and revisions and additions to FTUs.

## Discussion

FTUs are the basic building blocks of organs. Their distinctive size (relative to diffusion distances) and physical arrangement of different cell types are key to enabling their corresponding physiologic function. In this paper, we described ontology-aligned visualizations for 22 FTUs in 10 organs interconnected by vasculature as published in the 6th release of the HRA. To our knowledge, this is the first time that comprehensive data about FTUs in major vital organs and the vascular pathways connecting the FTUs to the heart and each other have been systematically cataloged.

FTU illustrations are published in widely used, standard pixel file format (PNG) for inclusion in presentations or publications but also in vector formats (SVG and AI) to support editing using Inkscape or Adobe Illustrator. The SVG vector format also supports the creation of interactive user interfaces in which users can hover over an anatomical structure or cell type featured in the FTU illustration to retrieve details on demand (e.g., labels, ontology IDs, mean biomarker expression values). Crosswalk tables link the individual elements in the SVG illustrations to the corresponding terms in the Anatomical Structures, Cell Types, and Biomarkers (ASCT + B) tables in the HRA[3], see Standard Operating Procedure entitled "Authoring Crosswalk Tables Between Functional Tissue Unit (FTU) Illustrations and ASCT + B Tables[8] for detailed instructions on how to author and use a crosswalk table. The ASCT + B tables provide a standardized vocabulary for describing the components of the human body, linked to existing ontologies.

While there are several limitations of this initial version of the FTU catalog (data came from a variety of sources, the FTUs and blood vasculature pathways currently do not list anatomical variants), we are using the data for several HRA use cases. For example, we ran Kaggle

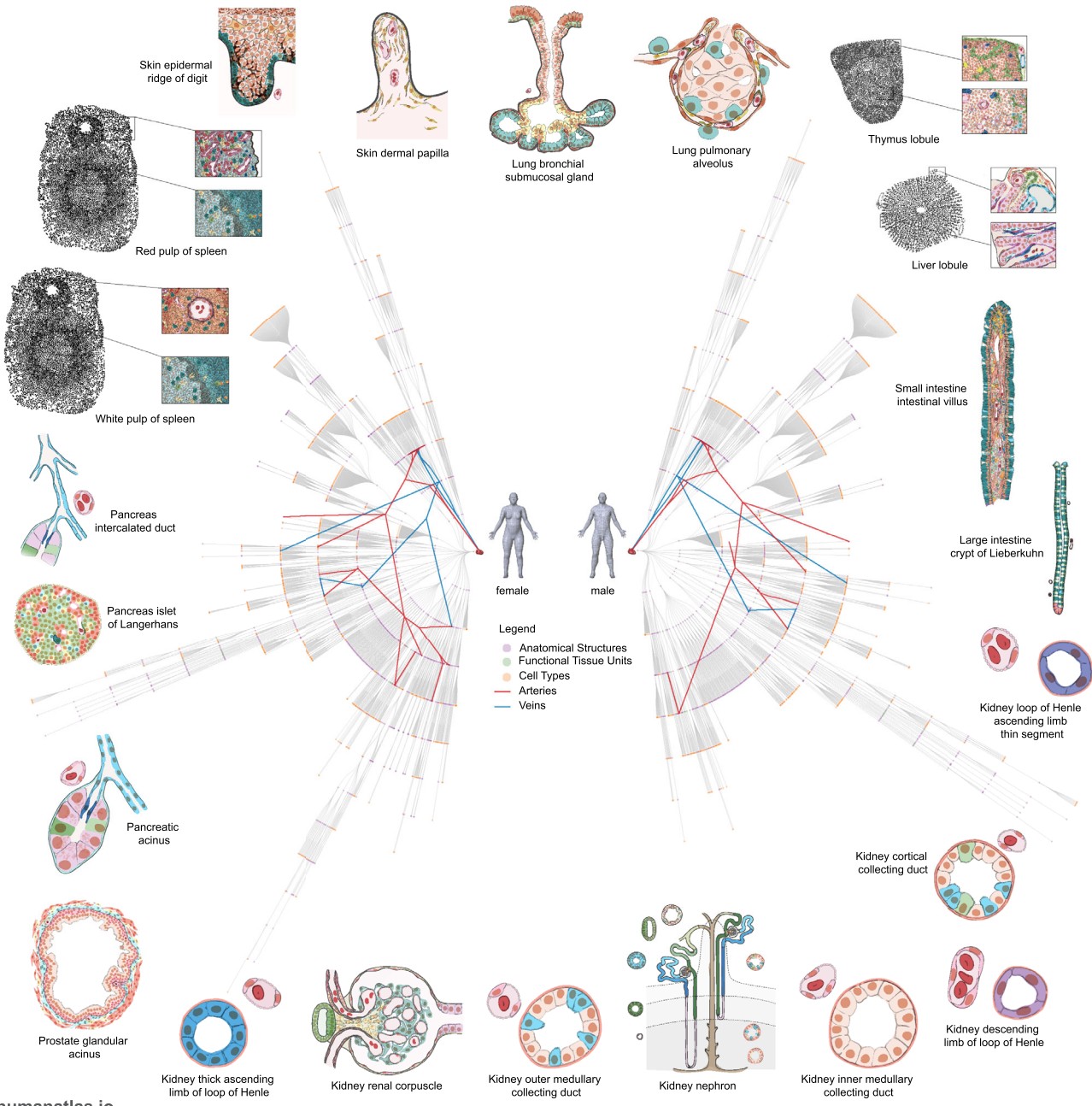

**Fig. 2 | HRA network.** Poster of the HRA illustrating radial tree graphs of (1) the nested partonomy of organ anatomical structures and cell types of the human body (female on left and male on right) with an overlay of (2) the branching structure of the blood vasculature extending from the heart (center of the figure) to FTUs (outer edge). Illustrations of all 22 FTUs are placed outside of the radial tree visualization for reference.

competitions to develop scalable and generalizable segmentation code to identify FTUs in images[2,9]. We use the segmentation code to automatically count the number of FTUs per unit area and to compute biomarker expression values for genes, proteins, lipids, and metabolites within FTUs. Ongoing research compares hierarchical cell neighborhoods[10] computed from tissue data with the 22 FTUs in the catalog presented here.

More broadly, we envision the FTU catalog as a framework for researchers to study how the information presented here (FTU dimensions, cell types, locations, etc.) varies across donor demographics (e.g., age, sex, race) and in different disease states. Understanding similarities and differences between FTUs can help predict adverse events of medications or suggest new drug targets. The data

and 2D illustrations can be used in education, especially for comparing the physiology and vasculature of different organs and tissues. The 6-foot (26" or 183 cm) diameter poster is freely downloadable and intended for general public outreach.

Although we do not yet have a complete map of all FTUs in the human body, these first 22 FTUs demonstrate how various data about FTUs, including physical dimensions, blood vasculature, cell types, and spatial orientation can be interlinked across scales, in support of creating a human reference atlas. Going forward, we will map experimental data from different data portals to the 22 FTUs and implement an interactive user interface that makes it possible to explore changes in cell type populations and biomarker expression values across different donor ages and disease states.

## Methods

HRA FTUs and experimental data visualizations were developed using an eight-step process: (1) consult the Anatomical Structure, Cell Types, and Biomarker (ASCT + B) tables to identify the anatomical structures and cell types present in an FTU; (2) identify FTU shape, dimensions, and cell types from experimental data on FTU geometries published in scholarly papers and histological visualizations; (3) a professional medical illustrator creates an initial pencil drawing of the FTU at the cellular level; (4) organ experts with extensive expertize in human anatomy and single-cell studies comment on the FTU properties and initial drawing; (5) a professional medical illustrator creates a vector-based drawing of the FTU guided by a Standard Operating Procedure (SOP) entitled "Creating 2D Reference Illustrations for FTU[11] and the "Style Guide for Human Reference Atlas 2D Functional Tissue Unit (FTU) Illustrations[12]; (6) organ experts review the drawings, metadata, and any existing disclaimers and suggest changes as needed which are implemented; (7) a crosswalk file is compiled that associates elements (anatomical structures and cell types) in the FTU vector file to their counterparts in the Anatomical Structures, Cell Types, and Biomarkers (ASCT + B) tables using the SOP titled "Authoring Crosswalk Tables Between Functional Tissue Unit (FTU) Illustrations and ASCT + B Tables[8]; and (8) the number of cells per cell type are recorded and the FTU 2D files is published with all metadata and the crosswalk via the HRA Portal as part of an HRA release.

All FTU illustrations have citable digital object identifiers (DOIs) and are freely available at the HRA Portal (https://humanatlas.io/2d-ftu-illustrations) together with a crosswalk file which maps the anatomy to the multispecies Uberon anatomy ontology and cell types to the Cell Ontology. FTUs are provided in Portable Network Graphic (PNG), Scalable Vector Graphics (SVG), and Adobe Illustrator (AI) formats.

### Reporting summary

Further information on research design is available in the Nature Portfolio Reporting Summary linked to this article.

## Data availability

All data is freely available via the HRA Portal at https://humanatlas.io and the GitHub repository at https://github.com/cns-iu/hra-ftu-vccf-supporting-information. The supplementary data can be found on Zenodo at https://doi.org/10.5281/zenodo.11477238 and on GitHub at https://github.com/cns-iu/hra-ftu-vccf-supporting-information/tree/main/data. Standard operating procedures are at https://humanatlas.io/standard-operating-procedures.

## Code availability

All code is freely available at https://github.com/cns-iu/hra-ftu-vccf-supporting-information and a snapshot of the code was published on Zenodo at https://doi.org/10.5281/zenodo.11477238.

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

## Acknowledgements

We thank organ experts Sanjay Jain, Matthias Kretzler, M. Todd Valerius (Kidney), John Hickey, Yiing Lin (Intestine), Gloria Pryhuber (Lung), Martha Campbell-Thompson (Pancreas), Douglas Strand (Prostate), Fiona Ginty, Arivarasan Karunamurthy (Skin), Andrea J. Radtke, Maigan Brusko (Spleen), Maigan Brusko, Andrea J. Radtke (Thymus), Anna Maria Masci, Tim Kendall, Ayako Suzuki (Liver) who supported FTU designs and revisions. Discussions with Becky Steck, Michael Rose, Todd M. Valerius, and Matthias Kretzler from the KPMP team who worked on the Kidney Tissue Atlas Explorer and Silvie Fexova, and Irene Papatheodorou from EBI who worked on the Anatomogram Explorer, informed the design of the HRA Portal (https://humanatlas.io). Heidi Schlehlein and Tracey L. Theriault designed the butterfly visualization poster using files provided by the authors of this paper. This research has been funded by the NIH Common Fund through the Office of Strategic Coordination/Office of the NIH Director under awards OT2OD033756 and OT2OD026671, by the Cellular Senescence Network (SenNet) Consortium through the Consortium Organization and Data Coordinating Center (CODCC) under award number U24CA268108, by the Kidney Precision Medicine Project grant U2CDK114886, by the NIDDK under awards U24DK135157 and U01DK133090 and by The CIFAR MacMillan Multiscale Human project. The funders had no role in study design, data collection and analysis, decision to publish, or preparation of the manuscript.

## Author contributions

S.B. compiled FTU data; K.B. lead this work as part of the HRA effort; E.Q. helped assign CL IDs to cell types in experimental data; M.N. and R.M. implemented the butterfly visualization; K.S.G. and G.M.W. compiled the vasculature data; R.B. designed the FTU illustrations with expert input by organ experts; E.M., B.H., and K.B. implemented the HRA Portal (https://humanatlas.io/2d-ftu-illustrations); S.B., M.N., K.B., and G.M.W. wrote the manuscript. All authors edited the manuscript.

## Competing interests

The authors declare no competing interests.
