## [Peer Review File · Nature Communications]

Reviewers' comments:

Reviewer #1 (Remarks to the Author):

This paper provides documentation and web-links to data on 'Functional Tissue Units' (FTUs), that represent the biological anatomical structures at which physiological mechanisms emerge from molecular and cellular biology. The data is being generated as part of an NIH Common Fund project. In some cases the FTUs are broken down into subcomponents (such as segments of the kidney nephron). The 22 FTUs described in the paper are documented in the links to a github repository which provides multiple ways of viewing the data, including 2D diagrams showing the cell types and their geometric arrangement in the FTU, and a link with the vascular connections to that FTU. Genetic, protein and lipid data will also be available for the cell types (although far from complete, a number currently have single cell gene expression data). Links are made to UBERON IDs for the anatomical structures and to a cell type ontology for the cells. Provenance is provided for data made available through the portal.

The FTUs described in the paper (from kidney, intestine, colon, liver, lung, pancreas, prostate, skin, spleen and thymus) represent a small fraction of all FTUs, but this paper documents the strategy being developed for the Human Reference Atlas (HFA) and provides a valuable resource that will expand as new data on FTUs becomes available.

A couple of very minor points:

- I suggest that 'anatomically based' is preferable to 'anatomically correct' in the abstract, since there will always be questions as to the anatomical accuracy.

- Under 'Geometric properties of FTUs', there is reference to 'circular-shaped FTUs' which I think should be 'spherically-shaped FTUs', since one dimension only (diameter) is used to describe the FTU, which is of course a 3D object.

Reviewer #1 (Remarks on code availability):

Very useful github repository.

Reviewer #2 (Remarks to the Author):

Summary and conclusions:

The paper and the Web tool it describes provide some useful annotated, interactive 2D images of 'functional tissue units' along with links to ontology terms and curation of FTU dimensions and blood supply from the literature. However, the bioinformatics analysis of 'biomarker gene expression' is relatively crude, very limited in scope and its presentation in the web tool is potentially misleading. The paper fails to detail the assumptions behind this analysis, its potential limitations and anticipated use cases and lacks any comparison to related efforts.

As currently written, I cannot recommend this paper for publication.

Review

The authors describe a web tool, the FTU explorer, that integrates interactive 2D schematics of 22 "functional tissue units" (FTUs) with cell counts and calculated "Gene Biomarkers" from selected single cell transcriptomics experiments. The 2D schematics include scale bars recording approximate dimensions derived from literature references. More details of dimensions along with supporting references are provided in a table in the paper. The paper also includes curated information about which blood vessels directly supply or drain each FTU and ontology mappings for cell types and tissues in the FTUs. Neither of these are available in the FTU explorer. The FTU explorer also includes slots for "Protein Biomarkers" and "Lipid Biomarkers" which are currently empty. Plans for populating these are not discussed in the text.

The cells for which "Gene Biomarkers" are provided in the FTU explorer are limited to those for which direct ontology mappings corresponding to ontology mappings in the curated FTU are available in the source dataset chosen in each case. "Gene Biomarkers" correspond to the 100 most highly expressed genes, where expression is calculated as mean expression across a cluster. They are presented alphabetical order in a table where each row corresponds to a cell type, so the tables have many more than 100 columns and a blank entry in any one cell indicates that the gene was below the top 100 cutoff for that cell, rather than that no expression was present. Expression is presented using colored circles with a color scale labelled "Expression Mean in FTU" and circle size labelled as representing "Percentage of Cells in FTU".

General Comments

The paper lacks any discussion of what constitutes a (useful) biomarker and fails to give any account of how the FTU explorer will help biologists to find useful markers for their work.

Typically, genes are referred to as markers* of cell types or tissues if their expression can be used to distinguish some cell type or tissue from others in the same tissue. Listing genes with the highest

levels of expression in alphabetical order does not seem like a good way to achieve this. Listing them side by side with gene expression for other cell type in the FTU may help, but given the use of a top 100 gene may also be misleading - it is possible that the lack of an entry merely indicates that expression was just below the cutoff. It is also not clear to me why mean expression levels across a set of cell sharing a cell type annotation can be described as "Expression Mean in FTU". The concept of FTU here is more obfuscating than enlightening.

* The term 'biomarker' is more commonly used in the biomedical domain where it typically denotes some indicator of a disease state, it is not clear what the use of the term biomarker adds in the context other than a connotation of clinical relevance.

The ASCT+B tables published by the same authors include expert provided marker list. Experts were explicitly asked for markers that "biomarkers that characterize and distinguish cell types". An analysis of whether these marker can be validated by single cell expression data would be a valuable addition to the field.

The meaning of circle size in the 'Gene biomarker' display is frankly baffling. This is normally used to indicate the percentage of expressing cells within an annotated cell set. According to the "FTU Explorer" site it indicates "Percentage of Cells in FTU" which a pop-up defined as "The percentage of cells in the FTU is calculated by dividing the total number of cells in all FTUs by the number of all cells in that tissue section." Looking at the analysis notebook on the linked GitHub repository does not clarify as the analysis appears to be loaded from a file on someone's hard drive.

What does "the total number of cell in all FTUs" refer to? Is it simply the number of cells of that type in the tissue? In which case, why mention FTUs at all? Any why is such a representation useful? Would it not merely indicate how rare a cell type is? This seems a very odd thing to bundle with presentation of expression.

There is no mention of how datasets were chosen and whether there are plans to aggregate. The analysis of "marker" expression feels to me like a poor copy of the much more comprehensive effort to perform gene expression analysis and provide tools to explore it on the CellXGene platform - the only difference being the context of 2D schematic browser. Plans for scaling are not discussed and the analysis pipelines presented as linked Jupyter notebooks are not scaleable.

Required changes

1. The text should include discussion of biomarkers - concepts, use cases, approach taken to calculating them & limitations of approach.
2. A reasonable justification of the current visualisation of the use of circle size in marker expression OR reversion to a more standard use of this to represent the percentage of annotated cells expressing the gene in question
3. It should be possible to run linked Jupyter notebooks. This would require a repo with library dependencies recorded in standard form and resolvable data dependencies referenced from the notebook (hosted either on the repo & referenced by local path or web resolvable and referenced as such.)

4. The paper should make clear the limited extent of bioinformatics analysis and the limited infrastructure currently developed (It would be more accurate to call this a pilot project). It should include a realistic assessment of how the work will scale. Alternatively, the authors might consider abandoning their current approach - perhaps re-using openly available analyses - adding value with cross analysis (e.g. do ASCT+B markers validate) + additional context.

5. Much supporting data is provided as links to tables on GitHub when it could be made available for download directly from the FTU explorer. Examples...

Specific Comments on text:

Line 45-6: "Here, we extend this definition of FTUs by allowing them to form a nested hierarchy to accomplish various functions."

The idea of nesting FTUs seems incompatible with a definition as "the smallest tissue organization that performs a unique physiologic function". The only nesting I see in the paper or in the FTU explorer is for nephron. I don't see what's gained by treating the larger composite unit as an FTU. What are the various function

s nesting accomplishes? I see no reference to this later in the paper?

Line 120-1: "the FTU Explorer makes it possible to examine bulk and spatial data"

In what sense does it include spatial "data" rather than just spatial schematics? This should be made explicit.

Line 124-6: "If experimental datasets for this FTU exist, cell types and biomarker expression values are tabulated"

The authors have not extensively assessed whether datasets for each FTU exist and incorporated them into the FTU explorer. There is also no discussion of analysis across aggregated datasets covering the same 'FTUs'. At best this should be described as a pilot effort, with details provided about how it will be scaled up.

Line 151 "diameter"

Is the poster circular?

Line 210-229 : section "Calculating Gene Expressions Per Cell Type"

This section is heavy on details of the standard libraries used, but does not communicate what we calculated. I had to inspect the linked Jupyter notebooks to work out that 'Gene Biomakers' correspond to the 100 most highly expressed genes. The text should be updated to make this clear.

What's missing:

- variation due to variation in quality of annotation.

Response to reviewer comments

We thank the reviewers for their helpful feedback and suggestions. Below is a point-by-point response to their comments.

Reviewer #1

- I suggest that 'anatomically based' is preferable to 'anatomically correct' in the abstract, since there will always be questions as to the anatomical accuracy.

We have made this suggestion and changed “anatomically correct” to “anatomically based”.

- Under 'Geometric properties of FTUs', there is reference to 'circular-shaped FTUs' which I think should be 'spherically- shaped FTUs', since one dimension only (diameter) is used to describe the FTU, which is of course a 3D object.

We have changed “circular-shaped” to “spherically-shaped”.

Reviewer #2

The paper and the Web tool it describes provide some useful annotated, interactive 2D images of 'functional tissue units' along with links to ontology terms and curation of FTU dimensions and blood supply from the literature. However, the bioinformatics analysis of 'biomarker gene expression' is relatively crude, very limited in scope and its presentation in the web tool is potentially misleading. The paper fails to detail the assumptions behind this analysis, its potential limitations and anticipated use cases and lacks any comparison to related efforts.

The main focus of this paper is supposed to be a catalog of annotated and linked information about FTUs. We included the web tool as a demonstration of how users might interact with this catalog. However, we acknowledge that this website was more of a prototype than a finished product and could be a distraction from the rest of the text. We have removed it from the paper to make the paper more concise and to limit the scope of the paper to the biological characteristics of the FTUs. We have also removed the biomarker discussion from the paper since this was only a component of the web tool and not fully described in this paper.

The authors describe a web tool, the FTU explorer, that integrates interactive 2D schematics of 22 "functional tissue units" (FTUs) with cell counts and calculated "Gene Biomarkers" from selected single cell transcriptomics experiments. The 2D schematics include scale bars recording approximate dimensions derived from literature references.

More details of dimensions along with supporting references are provided in a table in the paper. The paper also includes curated information about which blood vessels directly supply or drain each FTU and ontology mappings for cell types and tissues in the FTUs. Neither of these are available in the FTU explorer. The FTU explorer also includes slots for "Protein Biomarkers" and "Lipid Biomarkers" which are currently empty. Plans for populating these are not discussed in the text.

As noted above, we removed the FTU Explorer and discussion of biomarkers from the paper.

The paper lacks any discussion of what constitutes a (useful) biomarker and fails to give any account of how the FTU explorer will help biologists to find useful markers for their work.

This was not meant to be the main focus of this paper. We have removed the FTU Explorer and discussion about biomarkers.

1. The text should include discussion of biomarkers - concepts, use cases, approach taken to calculating them & limitations of approach.

We have removed biomarkers from this paper to ensure the focus is on the novel FTU catalog.

2. A reasonable justification of the current visualisation of the use of circle size in marker expression OR reversion to a more standard use of this to represent the percentage of annotated cells expressing the gene in question

We have removed the FTU Explorer web tool. This was a prototype and contained numerous limitations, as pointed out by the reviewer.

3. It should be possible to run linked Jupyter notebooks. This would require a repo with library dependencies recorded in standard form and resolvable data dependencies referenced from the notebook (hosted either on the repo & referenced by local path or web resolvable and referenced as such.)

We have removed the FTU Explorer relevant code from GitHub.

4. The paper should make clear the limited extent of bioinformatics analysis and the limited infrastructure currently developed (It would be more accurate to call this a pilot project). It should include a realistic assesment of how the work will scale. Alternatively, the authors might consider abandoning their current approach - perhaps re-using openly available analyses - adding value with cross analysis (e.g. do ASCT+B markers validate) + additional context.

We agree with the reviewer. We went a step further and removed the FTU Explorer web tool and biomarker discussion from the paper, since those were not meant to be the focus of this work.

5. Much supporting data is provided as links to tables on GitHub when it could be made available for download directly from the FTU explorer. Examples...

Without the FTU Explorer, the HRA Portal (<http://humanatlas.io/2d-ftu-illustrations>) and GitHub are our main way of making the FTU catalog available.

Line 45-6: "Here, we extend this definition of FTUs by allowing them to form a nested hierarchy to accomplish various functions."

The idea of nesting FTUs seems incompatible with a definition as "the smallest tissue organization that performs a unique physiologic function". The only nesting I see in the paper or in the FTU explorer is for nephron. I don't see what's gained by treating the larger composite unit as an FTU. What are the various functions nesting accomplishes? I see no reference to this later in the paper?

We expanded the text on nested FTUs in three places in the paper:

- 1) In the paragraph introducing the concept of nested FTUs, we added the following text to explain the rationale for nested FTUs: "This is needed in cases where multiple FTUs work together to accomplish complex functions that the individual FTUs cannot perform on their own. Just like the individual FTUs, these higher-level groups of FTUs are replicated many times."
- 2) In the next paragraph, after "Specifically, we present a catalog of the first 22 FTUs for the HRA", we added the text "including the kidney nephron as an example of a nested FTU made of smaller FTU structures."
- 3) In the Results section titled "HRA FTU Data", we added the text "Each FTU component of the larger nephron FTU performs a specific independent filtering step, but all are needed to perform the overall function of the nephron."

**Line 120-1: "the FTU Explorer makes it possible to examine bulk and spatial data"
In what sense does it include spatial "data" rather than just spatial schematics? This should be made explicit.**

We removed the FTU Explorer from the paper.

Line 124-6: "If experimental datasets for this FTU exist, cell types and biomarker expression values are tabulated"

The authors have not extensively assessed whether datasets for each FTU exist and incorporated them into the FTU explorer. There is also no discussion of analysis across aggregated datasets covering the same 'FTUs'. At best this should be described as a pilot effort, with details provided about how it will be scaled up.

We removed the FTU Explorer from the paper.

Line 151 "diameter"

Is the poster circular?

Yes, the illustration is circular.

Line 210-229 : section "Calculating Gene Expressions Per Cell Type"

This section is heavy on details of the standard libraries used, but does not communicate what we calculated. I had to inspect the linked Jupyter notebooks to work out that 'Gene Biomakers' correspond to the 100 most highly expressed genes. The text should be updated to make this clear.

This was removed from the paper along with the FTU Explorer website where this was used.

What's missing:

• variation due to variation in quality of annotation.

We added a paragraph on limitations to the Discussion: "There are several limitations of this initial version of the FTU catalog. The data came from a variety of sources, which may vary in quality or accuracy; the blood vasculature pathways currently do not list anatomical variants; and, the 2D illustrations represent a prototypical example of each FTU, but do not show the range in possible sizes and cell arrangements."

REVIEWER COMMENTS

Reviewer #2 (Remarks to the Author):

The authors have addressed critiques of the FTU explorer marker calculation and display by removing mention of the FTU explorer completely. They have also addressed the more minor specific issues with the text. Removing the FTU explorer and the associated marker analysis leaves only their work documenting FTUs via annotated images. As a result, the paper is now rather thin on content for a Nature Comms paper. I think it is for an editor to decide if it is sufficient for publication. However, these openly available, annotated FTU images are of value to the community. They are potentially useful to multiple atlasing and ontology browsing resources as a source of *interactive* images for browsing and selecting content by floating over or selecting parts of the image. The FTU explorer (now cut from the paper) illustrates how this can be achieved, as do the anamograms used in by the EBI Single Cell Expression Atlas. This is a valuable way to make ontology-linked atlas resources more easily browsable and accessible – as it makes the meaning of the ontology terms used in annotation explicit to biologists in a way that is directly coupled to the browsing mechanism and is much clearer than text.

However, it is not clear either from the paper or the linked resources how building these interactive resources might be possible using the files described and available via the HRA Portal. Downloads are available in various formats (svg, png, ai) along with general metadata for each image and a 'crosswalk table'. The most obvious format that might support re-use in interactive web is svg. I inspected these files, but there is no obvious way to find which layers/objects in the file correspond to which annotations, and it is not obvious (to me at least) what, if anything, in the crosswalk table might be used as a key to link annotations in the svg to ontology terms.

Required change:

- The paper or linked resources should include a guide to how to use these images to build interactive resources with explicit reference to the mappings between objects/layers in the SVG and ontology terms (cross walk table?) that make this possible.

Optional:

- The authors already have web solutions to this problem implemented in the FTU explorer. Publication of the javascript that drives this as a generic widget would greatly facilitate re-use, which I would argue is strongly in the author's interests.

Reviewer #2 (Remarks on code availability):

The code is relatively straightforward to follow and documentation is good (except for issues listed below). Some small changes are required to make this more usable:

1. Add a requirements.txt (or some equivalent) to specify library dependencies.
2. The authors should make a GitHub release with the version of code referred to in the paper and include a link to that release in the paper.
3. The specific ASCT+B JSON download file required to run the notebook* in the code folder should be documented more explicitly. There are a couple of options for this: (i) use an explicit request (e.g. get) in the code to download the file directly OR (ii) Add specific details of the file to download in the README in the code folder as well as in comments in the notebook.

* https://github.com/cns-iu/hra-ftu-vccf-supporting-information/blob/main/code/HRA_Butterfly_viz.ipynb

Reviewer #3 (Remarks to the Author):

The authors present a comprehensive and well-organized catalog of 22 functional tissue units (FTUs) across 10 human organs, providing key information about their physical dimensions, vasculature, cell types, and spatial arrangements. Mapping the anatomy terms to the Uberon and Cell Ontology standards facilitates computational access and integration with other datasets. The nested hierarchy of FTUs is a novel and useful extension of the FTU concept that enables modeling of complex functions.

This work makes a valuable contribution to the ongoing development of the Human Reference Atlas. By systematically compiling data on FTUs that was previously scattered across the literature, the authors have created a framework for comparing FTUs across the body and studying how their characteristics vary with demographics and disease states. The illustrations, datasets, and poster

will be highly useful resources for research and education. Although the catalog is not yet complete, this initial set of 22 FTUs demonstrates the feasibility and value of the approach.

In summary, this paper reports the development of a unique and extensive FTU catalog that will facilitate research, modeling, and education related to human anatomy and physiology. The work is comprehensive, rigorous, well-presented, and will be of broad interest. I strongly support its publication in Nature Communications.

Response to reviewer comments

We thank the reviewers for their helpful feedback and suggestions. Below is a point-by-point response to their comments.

Reviewer #2

The authors have addressed critiques of the FTU explorer marker calculation and display by removing mention of the FTU explorer completely. They have also addressed the more minor specific issues with the text. Removing the FTU explorer and the associated marker analysis leaves only their work documenting FTUs via annotated images. As a result, the paper is now rather thin on content for a Nature Comms paper. I think it is for an editor to decide if it is sufficient for publication. However, these openly available, annotated FTU images are of value to the community. They are potentially useful to multiple atlas and ontology browsing resources as a source of *interactive* images for browsing and selecting content by floating over or selecting parts of the image. The FTU explorer (now cut from the paper) illustrates how this can be achieved, as do the anatomograms used in by the EBI Single Cell Expression Atlas. This is a valuable way to make ontology-linked atlas resources more easily browsable and accessible – as it makes the meaning of the ontology terms used in annotation explicit to biologists in a way that is directly coupled to the browsing mechanism and is much clearer than text.

However, it is not clear either from the paper or the linked resources how building these interactive resources might be possible using the files described and available via the HRA Portal. Downloads are available in various formats (svg, png, ai) along with general metadata for each image and a 'crosswalk table'. The most obvious format that might support re-use in interactive web is svg. I inspected these files, but there is no obvious way to find which layers/objects in the file correspond to which annotations, and it is not obvious (to me at least) what, if anything, in the crosswalk table might be used as a key to link annotations in the svg to ontology terms.

Required change:

- The paper or linked resources should include a guide to how to use these images to build interactive resources with explicit reference to the mappings between objects/layers in the SVG and ontology terms (cross walk table?) that make this possible.

Authors: Excellent suggestion. We expanded the Discussion and Methods sections and added a link to a detailed Standard Operating Procedure (SOP) entitled “Mapping Functional Tissue Unit (FTU) Illustrations to ASCT+B Tables” (<https://zenodo.org/records/10359003>) with information on how the two-dimensional SVG object layers are mapped to the ontology terms in the HRA.

Optional:

- The authors already have web solutions to this problem implemented in the FTU explorer. Publication of the javascript that drives this as a generic widget would greatly facilitate re-use, which I would argue is strongly in the author’s interests.

Authors: All HRA code is publicly available on GitHub and at <https://github.com/hubmapconsortium/hra-ui>. The FTU Explorer is under active development (and will be detailed in a separate paper) and there will be lightweight integration components for interactive FTU visualizations that others can use in their data portals or other websites.

Reviewer #2 (Remarks on code availability):

The code is relatively straightforward to follow and documentation is good (except for issues listed below). Some small changes are required to make this more usable:

1. Add a requirements.txt (or some equivalent) to specify library dependencies.

Authors: We added a requirements.txt file to the `/code` directory on GitHub (<https://github.com/cns-iu/hra-ftu-vccf-supporting-information>) that provides version numbers for all software packages needed to run the code.

2. The authors should make a GitHub release with the version of code referred to in the paper and include a link to that release in the paper.

Authors: The static version of the data and code can be found in Zenodo under the DOI: [10.5281/zenodo.11477238](https://doi.org/10.5281/zenodo.11477238).

3. The specific ASCT+B JSON download file required to run the notebook* in the code folder should be documented more explicitly. There are a couple of options for this: (i) use an explicit request (e.g. get) in the code to download the file directly OR (ii) Add

specific details of the file to download in the README in the code folder as well as in comments in the notebook.

Authors: *We updated the README file in the /code directory to provide instructions for downloading the ASCT+B JSON file. We also provide all ASCT+B JSON files from HRA version 1.4 that are used to generate the butterfly visualization in the /code/data/v14 directory.*

Reviewer #3 (Remarks to the Author):

The authors present a comprehensive and well-organized catalog of 22 functional tissue units (FTUs) across 10 human organs, providing key information about their physical dimensions, vasculature, cell types, and spatial arrangements. Mapping the anatomy terms to the Uberon and Cell Ontology standards facilitates computational access and integration with other datasets. The nested hierarchy of FTUs is a novel and useful extension of the FTU concept that enables modeling of complex functions.

This work makes a valuable contribution to the ongoing development of the Human Reference Atlas. By systematically compiling data on FTUs that was previously scattered across the literature, the authors have created a framework for comparing FTUs across the body and studying how their characteristics vary with demographics and disease states. The illustrations, datasets, and poster will be highly useful resources for research and education. Although the catalog is not yet complete, this initial set of 22 FTUs demonstrates the feasibility and value of the approach.

In summary, this paper reports the development of a unique and extensive FTU catalog that will facilitate research, modeling, and education related to human anatomy and physiology. The work is comprehensive, rigorous, well-presented, and will be of broad interest. I strongly support its publication in Nature Communications.

Authors: *Thank you for your expert suggestions, deeply appreciated.*

REVIEWERS' COMMENTS

Reviewer #2 (Remarks to the Author):

I am satisfied the the authors have made all the required changes requested. More specifically, the documentation provided and the linked files are now sufficient for consumers of these annotated images to use them to build ontology-linked interactive web resources. The authors have also updated the linked data and code sufficiently to support re-usability. In short, I am now happy to approve publication.

Reviewer #2 (Remarks on code availability):

The authors have updated the linked data and code sufficiently to support re-usability.